# Can Language Models Learn to Skip Steps?

**Tengxiao Liu**♠▲*   **Qipeng Guo**♡   **Xiangkun Hu**◇   **Cheng Jiayang**◇
**Yue Zhang**♣†   **Xipeng Qiu**♠†   **Zheng Zhang**◇
♠Fudan University   ▲UC Santa Barbara   ♡Shanghai AI Laboratory
♣Westlake University   ◇Amazon AWS AI
tengxiao@ucsb.edu   zhangyue@westlake.edu.cn
xpqiu@fudan.edu.cn   zhaz@amazon.com

## Abstract

Trained on vast corpora of human language, language models demonstrate emergent human-like reasoning abilities. Yet they are still far from true intelligence, which opens up intriguing opportunities to explore the parallels of humans and model behaviors. In this work, we study the ability to skip steps in reasoning—a hallmark of human expertise developed through practice. Unlike humans, who may skip steps to enhance efficiency or to reduce cognitive load, models do not inherently possess such motivations to minimize reasoning steps. To address this, we introduce a controlled framework that stimulates step-skipping behavior by iteratively refining models to generate shorter and accurate reasoning paths. Empirical results indicate that models can develop the step skipping ability under our guidance. Moreover, after fine-tuning on expanded datasets that include both complete and skipped reasoning sequences, the models can not only resolve tasks with increased efficiency without sacrificing accuracy, but also exhibit comparable and even enhanced generalization capabilities in out-of-domain scenarios. Our work presents the first exploration into human-like step-skipping ability and provides fresh perspectives on how such cognitive abilities can benefit AI models.

## 1   Introduction

The pursuit of Artificial General Intelligence (AGI) is profoundly influenced and inspired by human intelligence [35, 6]. Trained extensively on human language, language models not only excel in various tasks, but also begin to exhibit emergent human-like abilities that are not explicitly engineered into them [24]. Among these, reasoning stands out as a core human-like cognitive ability, and has demonstrated great potential in a wide range of problem solving scenarios [47, 11, 30, 37, 28, 34]. Despite their advances in displaying human-like cognitive activities, huge gaps remain in how models and humans actually behave [22, 46, 20]. These differences bring up interesting questions regarding the exploration and development of similar capabilities between models and humans.

We aim to investigate whether the models exhibit any reasoning abilities unique to human experts, and whether they can evolve from beginners to reasoning experts. When humans learn to reason, beginners typically start with detailed, step-by-step solutions to imitate the gradual process of problem solving. As practice makes perfect, human experts not only solve problems more swiftly but also utilize shorter mental pathways, often skipping steps in their reasoning process [36]. This particular ability helps them speed up the reasoning and saves cognitive load for more challenging steps [44]. As demonstrated in Figure 1, the step-skipping behavior illustrated on the right side is commonly adopted by human experts during equation simplification.

---

*Work done during internship at AWS Shanghai AI Lab.
†Corresponding authors.

38th Conference on Neural Information Processing Systems (NeurIPS 2024).

```
Q: A * B * x * C – D / F = 0        Q: A * B * x * C – D / F = 0
Solve it in 4 steps.                Solve it in 2 steps.

Step1: A * B * x * C = D / F         Step1: A * B * x * C = D / F
Step2: A * B * x = D / F / C         (Skip) A * B * x = D / F / C
Step3: B * x = D / F / C / A         (Skip) B * x = D / F / C / A
Step4: x = D / F / C / A / B         Step2: x = D / F / C / A / B
```

Figure 1: Step skipping in equation simplification. We use the specified number of steps in the input as a stimulation to induce the model to perform skipping by using fewer steps.

In this work, we are curious whether models exhibit mature human-like reasoning ability — skipping steps, and how such abilities can influence the model's reasoning behaviors. Unlike humans, models do not inherently possess the intrinsic motivation like time limit or skill maturity that naturally drives efficiency in cognitive tasks. To induce the skipping step behavior in models, we introduce a controlled training environment where models are instructed to generate reasoning sequences within a specified number of steps. Our method includes two phases: *initialization* and *iteration*. We begin with a dataset that contains complete stepwise reasoning processes for the questions. In initialization, models are first trained to solve the tasks comprehensively, adhering to the full sequence of reasoning steps. In Figure 1, the illustration on the left demonstrates how models are trained to follow a specified number of steps. Then in the iteration phase, the models are prompted to produce shorter answers based on the original training data (Figure 1 right). We then select the shorter reasoning paths that still achieve correct answers and mix them with the full-step reasoning paths. This expanded dataset is used to train a new model to have advanced step-skipping capabilities. Each iteration refines the model's ability to identify how steps can be skipped without sacrificing accuracy. Finally, we fine-tune the models using these iteratively generated datasets, including data instances that demonstrate successful step-skipping during each iteration.

We conduct experiments with three different reasoning datasets, each characterized by clear internal reasoning steps, to evaluate model behaviors. Empirical results demonstrate that models exhibit and develop the ability of skipping steps in our framework - not only solving tasks effectively but also actively omitting steps to enhance efficiency. Further analysis of model behaviors indicate that these skipped reasoning paths act as beneficial enhancements rather than mere biased shortcuts, as evidenced by their maintenance or even improvement of out-of-distribution (OOD) performance across various tasks. To the best of our knowledge, this work is the first investigation into the human-like ability of step-skipping in language models, providing empirical evidence that models can indeed skip steps. These preliminary findings provide a fresh perspective on easy-to-hard generalization — training models on simpler data comprising both comprehensive and skipped reasoning steps can enhance their ability to generalize to more complex scenarios. [¶]

## 2   Related Work

**Human-like Abilities in Language Models**   Many of the capabilities widely used in current models are inspired by human intelligence. For instance, in-context learning enables models to address problems by mimicking the patterns demonstrated in examples [5]. In reasoning tasks, models benefit from progressively answer derivations and step-by-step chain-of-thought processes [47] and their humanlike enhancements, such as planning [18], task decomposition [50], and refinement [32, 38]. Another series of studies explore from the perspectives of cognitive science and psychology [10, 2, 12, 9]. Kosinski [24] reveal that current large language models have demonstrated a certain level of Theory-of-Mind (ToM) abilities by testing their performance to impute another's mental states and perspectives. Further studies [21] provide preliminary evidence of a correlation between the embeddings in LLMs and human brain neurons during ToM tasks, while Ma et al. [31] highlights the limitation of current ToM evaluations as they target narrow and inadequate aspects of ToM. Apart from these cognitive abilities, our work draws inspiration from human problem solving [23, 42, 3, 44] and evaluates language models on these unique step skipping behaviors. Additionally, our work aligns with an expanding field exploring the correlation between System 1 and System 2 reasoning

---

[¶]Code and data are publicly available at: `https://github.com/tengxiaoliu/LM_skip`.

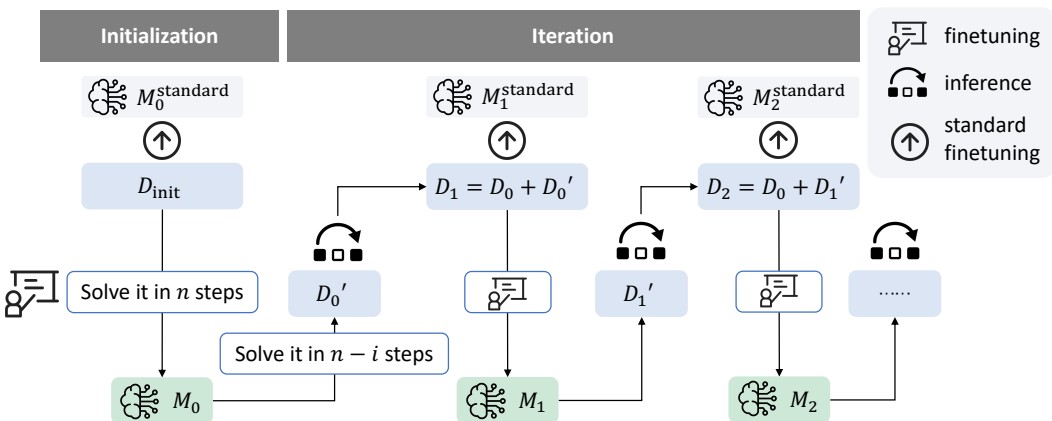

Figure 2: Overall framework. The initialization phase aims to equip the model with the ability to reason according to a specified number of steps. During iterations, each cycle produces a mixed dataset $D_i$, which is used to train a standard model to evaluate the model's step-skipping capabilities.

mechanisms [14, 15, 49]. Rather than removing all reasoning trajectories, our work explores gradual shortening to provide a smoother transition that mirrors natural cognitive processing.

**Compositional Generalization Challenges**   Transformers have shown limitations in complex compositional generalization scenarios [17, 39]. Previous work also indicates that models may develop biased shortcut, negatively impacting their OOD performance [27, 25, 16]. A growing body of research focuses on easy-to-hard generalization [4, 7, 19, 41, 48], where models improve their generalization ability by learning from easy tasks, without requiring intensive supervision on harder ones. Following this line, our work encourages the model to learn from self generated skipping paths, which has been empirically shown to maintain and even enhance OOD generalization capabilities.

## 3   Method

Humans develop the ability to skip steps for several reasons. With practice in specific tasks, they evolve from novices to experts, optimizing lengthy thought processes into quicker, more efficient reasoning. Additionally, factors such as time constraints or the desire to conserve cognitive resources can also prompt humans to skip steps [13]. In contrast, models lack an inherent cognitive signal that would drive them to minimize reasoning steps. Rather than attempting to replicate these human-like signals, we design a training approach to directly control the number of steps in their reasoning processes. By restricting the steps in model responses, we can guide the model to self-generate data including skipped steps. Our framework has two phases: initialization and iteration.

### 3.1   Initialization

We begin with a training dataset $D_0$, which contains detailed full-step reasoning answers to the questions. Our goal is to train a model that can generate answers by following the specified number of steps in the input question. Depending on the characteristics of different tasks, there are two design choices to initialize the framework: cold start and warm start.

**Cold start**   In the cold start approach, we directly fine-tune the model on the original full-step training data, i.e., $D_{\texttt{init}} = D_0$. The trained model is expected to not only learn to solve the problems, but also adhere to the specified number of steps in the input instructions.

**Warm start**   Training exclusively with full steps does not always guarantee the ability of controlling the number of steps, especially for the challenging tasks. Therefore, we manually create answers that contain skipped steps based on human expertise. Optionally, we can also randomly merge adjacent steps or simply omit random steps within the rationales to create such skipped-step data. In either way, we can expand the original training set with additional data $D_{\texttt{skip}}$ that can better help models

learn how to solve the problems with fewer steps. Thus, the data for warm start initialization can be describes as $D_{\texttt{init}} = D_0 + D_{\texttt{skip}}$.

Using the prepared data, we fine-tune the model to generate the answers with the given number of steps. For each QA pair in $D_{\texttt{init}}$, the question $q$ is concatenated with the instruction $I_n$ which indicates that the reasoning process $a^{(n)}$ should be completed in $n$ steps. Therefore, the resulting model in the initialization phase, $M_0$, is described as:

$$M_0 = \prod_{(q,a^{(n)}) \in D_0} P(a^{(n)}|q, I_n), \qquad (1)$$

where the instruction $I_n$ stands for "Solve it in $n$ steps".

## 3.2  Iteration

After the initialization, the model is expected to have learned to solve the problems with detailed steps using the specified number of steps in the input. Leveraging this particular ability, we can encourage the model to actively engage in step skipping behavior. At the beginning of each iteration $k$, the model $M_{k-1}$ is prompted to solve the same problems in the training set using fewer steps than the full number. Responses that are both correct and meet the reduced step criterion are filtered and composed into a new dataset $D'_k$. These reasoning answers are generated solely by the model itself, reflecting its understanding after training on the initialized data and demonstrating its active preferences when reducing steps.

We define the dataset used for current iteration as $D_k = D_0 \cup D'_{k-1}$, where the original training set $D_0$ includes full reasoning steps and the filtered dataset $D'_{k-1}$ contains new responses that successfully utilized fewer steps. This ensures that the model has access to both the original complete reasoning processes and examples of effective step-skipping generated by the model itself. To finalize current iteration, the model $M_k$ is trained on $D_k$: $M_k = \prod_{(q,a^{(n)}) \in D_k} P(a^{(n)}|q, I_n)$.

The iterative training process described above requires specifying the number of steps in the input, which is impractical in real-world applications because it can be difficult to determine the exact number of steps needed for a given question. To be more applicable, we aim to understand how models learn from the generated skipped data and what benefits they can derive from it. Therefore, for each intermediate resulting dataset $D_k$, we train a new model using a standard QA finetuning setting without specifying the number of steps in the input:

$$M_k^{\texttt{standard}} = \prod_{(q,a^{(n)}) \in D_k} P(a^{(n)}|q). \qquad (2)$$

This phase aims to solidify the model's skipping behavior, simulating a more advanced form of cognitive processing akin to human reasoning.

# 4  Experiments

## 4.1  Datasets

We design three tasks to investigate the model's step skipping behavior (Figure 3). In each task, the intermediate steps needed to solve these problems are explicitly detailed and well-defined, facilitating a clear analysis of the model's predictions. When creating skipped data for warm start, we either omit certain steps or heuristically merge two adjacent steps. Details on data creation can be found in Appendix B.1.

**Analog of Algebra**  Following Blessing and Anderson [3], we create an analog of algebra by replacing the variables and operators with different symbols. As shown in Figure 3, each variable and standard operator is mapped to a unique, unrelated symbol. The desired result is to isolate the symbol ♥ (i.e., $x$) on the left side of the symbol ↔ (i.e., =). This task is entirely new for the model, making it an ideal scenario to understand how models develop problem-solving abilities from scratch. We use a heuristic script to generate the questions along with the stepwise solutions. After generating the QA pairs, we filter the data based on the number of variables involved in the question and the steps required to solve it. The training and in-domain test data contains questions with up to

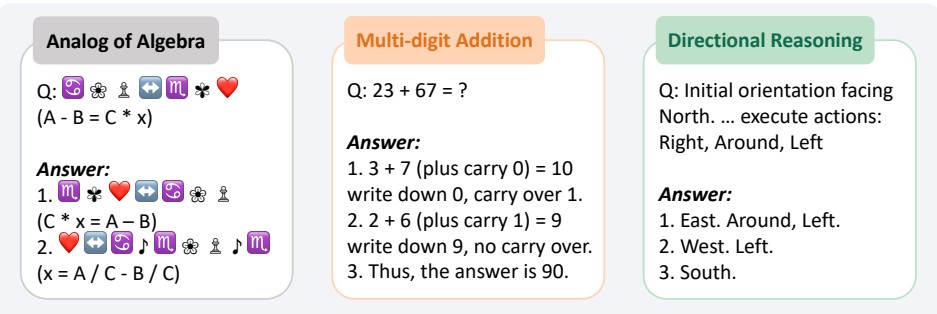

Figure 3: Illustrations of three different tasks. Each question is accompanied by a comprehensive detailed step-by-step solution.

7 variables and requiring no more than 5 steps. In addition, we create two out-of-domain datasets of varying difficulties to evaluate generalization performance: OOD-easy includes variables unseen during training, with 8 and 9 variables, no limit on steps. OOD-hard is the most challenging setting, including 10 - 14 variables and $\geq 9$ steps to solve. Both OOD sets contain unseen variables.

**Multi-digit Addition**   As a basic arithmetic task, multi-digit addition naturally involves detailed stepwise reasoning processes, serving as a suitable task for studying model behaviors in compositionality generalization[45, 26, 40]. We utilize step-by-step reasoning processes to perform addition operation digit by digit, as illustrated in Figure 3. For training and in-domain test data, we only consider additions involving numbers with up to 3 digits. We introduce two out-of-domain datasets depending on the number of digits involved in the addition: OOD-easy includes one number with up to 3 digits and another with 4-7 digits. OOD-hard contains two numbers, both with 4-7 digits.

**Directional Reasoning**   We additionally consider long-form symbolic directional reasoning, which poses a challenge for direct solution and necessitates continuous reasoning steps to arrive at the answer. This task provides an initial direction and a list of turning actions. The desired answer is the final facing direction. For training and in-domain test set, we consider questions that contain $\leq 10$ actions. OOD-easy includes questions with 11-20 actions and OOD-hard includes questions with 21-30 actions. The detailed statistics of three datasets can be found in Table 1.

Table 1: Dataset statistics.

| Task | Train | In-domain test | OOD-easy | OOD-hard |
|---|---|---|---|---|
| Analog of Algebra | 5,770 | 1,000 | 2,000 | 420 |
| Multi-digit Addition | 2,885 | 1,000 | 1,200 | 1,600 |
| Directional Reasoning | 2,080 | 1,000 | 500 | 500 |

## 4.2   Experiment Setting

For all our experiments, we use Llama 2 (7B parameters) [43] and phi-3-mini (3.8B parameters, with context length of 4K) [1] as our base model. We train the model using a learning rate of 5e-6 for 2 epochs with the AdamW optimizer [29]. During inference, we employ greedy decoding. We run our experiments with three different random seeds and report the average and standard deviation. All experiments are conducted on eight V100 GPUs each with 32GB memory. The total training time required to complete one full cycle of five iterations is under six hours.

## 5   Results

### 5.1   Can models learn to skip steps?

To make sure our framework can proceed to iterations smoothly, one crucial factor is the initialized model's ability to adhere to the specified number of steps in the input. In the cold start setting, we

Table 2: Step number following ability of the initialized Llama2 models across different tasks. "# Skipping" represents the number of instances where $n - i > 0$. "Step Consistency" quantifies the match between the actual number of steps taken and the number indicated in the input. "Answer Accuracy" calculates the percentage of correct final answers out of the "# Skipping" cases. "Average Step" reflects the mean number of steps across all predictions within the dataset.

| | Analogy of Algebra | | Multi-digit Addition | | Directional Reasoning | |
|---|---|---|---|---|---|---|
| | $i = 1$ | $i = 2$ | $i = 1$ | $i = 2$ | $i = 1$ | $i = 2$ |
| # Skipping | 5,308 | 4,159 | 2,844 | 2,175 | 2,071 | 2,049 |
| Step Consistency | 100.00 | 99.19 | 100.00 | 100.00 | 86.24 | 39.19 |
| Answer Accuracy | 8.14 | 2.77 | 98.35 | 82.58 | 85.47 | 29.62 |
| Average Step | 2.33 | 1.81 | 1.90 | 1.38 | 6.14 | 6.66 |

train the model exclusively using the full step training data. We then run inference on the training set, instructing the model to use $n - i$ steps to solve the question, where $n$ denotes the original full step number and $i \in [1, 2]$. If $n - i \leq 0$, we do not ask the model to try skipping on such cases and instruct the model to use $n$ steps instead.

As shown in Table 2, the results demonstrate that the fine-tuned model exhibits good step-number following ability on the Analog of Algebra — over 99 % of the answers follow the given number of steps. Additionally, when prompted to generate condensed answers with fewer steps, the model can produce some correct answers in the specified number of steps, achieving accuracies of 8.14% and 2.77% respectively. Despite this relatively low accuracy, these small amount of correct data can still assist the model in gradually developing step skipping ability through iterations. Ultimately, the model manages to produce over 90% of correct skipping data. The trend of the data quantity change can be found in Appendix B.2.

However, this ability varies across different tasks. For the other two tasks, models do not naturally develop the capability for active step skipping, leading to near zero step consistency when required to provide answers in fewer steps. To address this issue, we employ the warm start setting for these tasks. Table 2 presents the results of Multi-digit Addition and Directional Reasoning under the warm start setting, indicating that this approach enhances the models' proficiency with step skipping.

Ideally, we aim for models to be initialized through cold start. The benefits of this approach are obvious — it allows the model to spontaneously develop step skipping behavior, giving it sufficient freedom to decide and control which steps to skip. However, our experiments have revealed that it can be challenging for models to develop such capability in all scenarios. In contrast, the warm start offers an alternative design choice by providing human-created skipped data. This data includes intuitive and valid skipping steps derived from human expertise, making it more natural and helping models develop human-understandable behaviors. However, it might also introduce human biases that constrain the model's independent exploration of step skipping. This influence can be mitigated in the subsequent iteration phase, where the model is given full freedom to develop and amplify its own step-skipping behavior.

## 5.2 What do models learn from skipping steps?

Based on this new mixed data including both complete and skipped answers at each iteration, we train the standard models to analyze the change of model's performance — what models can learn from the behavior of skipping steps.

**Models learn to solve problems more effectively with fewer steps.** We evaluate the standard models on both in-domain and OOD data, with the results presented in Table 3. Detailed results from each iteration of the evaluation can be found in Appendix B.3. Given the simplicity of the tasks, the model is able to overfit on in-domain data, achieving nearly perfect performance. Further iterations of skipping steps manage to guide the model to use fewer steps while maintaining the performance. In two OOD scenarios, we find that the model trained with mixed data performs comparably to the model trained with complete steps on the OOD test sets, and even exhibits superior generalization abilities. Specifically, in Analog of Algebra, Llama2 models of iteration 5 achieves 4.76% gain on OOD-easy, while phi-3-mini achieves 7.08% gain on OOD-hard set. In the Multi-digit Addition task, the Llama2 model demonstrates a 13.91% improvement in OOD-easy performance

Table 3: Performance comparison of models from different phases. Avg steps denotes the average number of steps taken in the prediction. With the skipped step data, models achieve even better generalization performance with fewer steps.

| Task | Iteration | In-domain | | OOD-easy | | OOD-hard | |
|---|---|---|---|---|---|---|---|
| | | Acc | Avg steps | Acc | Avg steps | Acc | Avg steps |
| *Llama2-7B* | | | | | | | |
| Analog of Algebra | Cold start | 99.87 | 3.19 | 85.91 | 4.79 | 7.94 | 11.57 |
| | Iter 5 | 99.80 | 2.43 | **90.67** | 4.05 | **8.10** | 10.92 |
| Multi-digit Addition | Cold start | 100.0 | 2.86 | 0.06 | 3.25 | 0.00 | 3.69 |
| | Warm start | 99.53 | 2.72 | 0.14 | 3.02 | 0.11 | 3.49 |
| | Iter 5 | 99.17 | 1.46 | **13.97** | 1.49 | **4.75** | 2.06 |
| Directional Reasoning | Cold start | 100.0 | 7.01 | **90.00** | 15.77 | 42.00 | 19.39 |
| | Warm start | 99.97 | 6.28 | 87.20 | 14.65 | 42.33 | 18.02 |
| | Iter 5 | 100.0 | 6.45 | 89.33 | 14.87 | **51.80** | 19.49 |
| *phi-3-mini* | | | | | | | |
| Analog of Algebra | Cold start | 99.60 | 3.19 | 98.04 | 6.16 | 4.05 | 10.01 |
| | Iter 5 | 99.90 | 2.75 | **98.95** | 5.60 | **11.13** | 7.98 |
| Multi-digit Addition | Cold start | 99.92 | 2.86 | 35.93 | 5.03 | 5.39 | 5.44 |
| | Warm start | 99.97 | 2.62 | 39.08 | 3.80 | 5.11 | 4.06 |
| | Iter 5 | 99.93 | 2.08 | **46.61** | 2.31 | **14.98** | 2.59 |
| Directional Reasoning | Cold start | 99.83 | 7.01 | 91.47 | 15.46 | 62.67 | 24.85 |
| | Warm start | 99.80 | 6.82 | 93.67 | 15.19 | 71.80 | 24.61 |
| | Iter 5 | 99.70 | 6.12 | **93.73** | 14.44 | **73.87** | 23.77 |

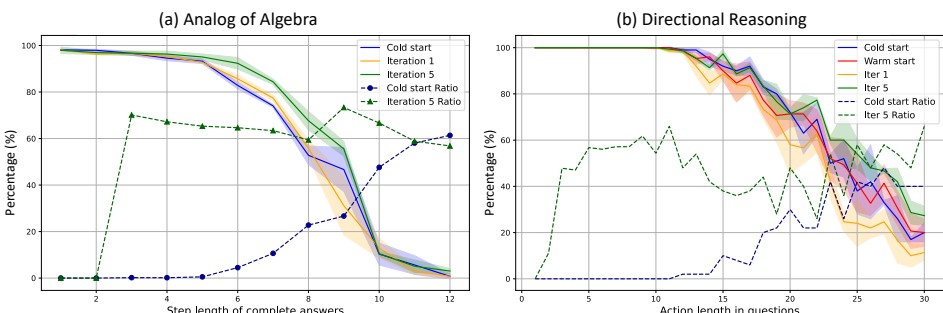

Figure 4: Comparison of models across different phases relative to question length and complexity. Models achieve near perfect performance on in-domain data but diverge on lengthy OOD data.

and a 4.75% increase in OOD-hard performance. In the OOD-hard dataset for Directional Reasoning, Llama2's performance improvs by 9.2%. These results suggest that not only is the model unaffected by potential shortcut bias from the skipping steps, but it actually benefits from the mixed training data to gain enhanced task solving abilities. The ablation analysis on various data-mixing approaches are provided in Appendix B.5. Furthermore, we observe that the model uses fewer steps, thereby increasing problem-solving efficiency.

## 5.3 Model Behavior Analysis

### 5.3.1 Analog of Algebra

Figure 4(a) presents the performance of Llama2 models across various iterations in the Analog of Algebra task, differentiated by the number of steps required in the complete answers. The solid lines represent the accuracy of final answers. We perform uniform evaluation on the union of all in-domain and OOD test sets. Initially, all models maintain high accuracy for in-domain problems with up to five steps, after which a significant drop is observed as the complexity increases. As the model undergoes

iterations, there is a noticeable improvement in its ability to handle longer step lengths (green solid line), particularly in the range of 6 to 10 steps where other models show significant weaknesses. The dashed lines illustrate the proportion of data exhibiting step-skipping in model predictions. The blue dashed line indicates models initially adopt step-skipping as problems extend in length. After iterations, the green dashed line indicates the models consistently employ step skipping in shorter questions, thereby improving the reasoning efficiency.

### 5.3.2 Directional Reasoning

Figure 4 (b) illustrates the comparison of Llama2 model's performance across different question lengths on Directional Reasoning task. We observe that the artificial skipped data has minimal impact on the model, with negligible differences between the cold start and warm start phases. Upon entering the iterative phase, the model's performance notably declines during the first iteration, particularly in handling longer problems. This downturn may reflect the model's adjustment from manually injected skipped data to its own step skipping ability. Subsequent iterations show that the model benefits more significantly from data generated during the iteration process, as evidenced by the results in Iteration 5. The model maintains consistency with the baseline in both in-domain and out-of-domain performances, and exhibits a slight advantage in solving longer problems. Similar to the previous task, the Iteration 5 Ratio curve (dashed green line) also shows a significant increase in step-skipping behavior, suggesting an evolved efficiency in reasoning as the model opts to bypass steps while maintaining or even improving accuracy.

### 5.3.3 Multi-digit Addition

In Figure 5, we show a finer-grained evaluation of multi-digit addition tasks on Llama2. The horizontal and vertical axes of the matrices represent the number of digits in the two addends for each question in the test set (both in-domain and OOD test data). We utilize the following three metrics: **Question-level accuracy** assesses whether the final answer is correct for additions involving different numbers of digits. **Step-level distribution** illustrates the distribution of the digit lengths used in each individual step of the model's stepwise reasoning process. **Step-level accuracy** measures the accuracy of the single step calculations involving different numbers of digits.

In Figure 5(a), as iterations progress, the model demonstrates improved generalization performance across all test datasets. When initialized with a cold start, the model can only learn from the training data involving single-digit addition steps, resulting in overfitting to in-domain test data (digit $\leq 3$). When augmented with manually created skipped data for a warm start, the model begins to incorporate multi-digit additions with skipped steps. However, the inconsistency between the manually injected data and the model's inherent behavior does not significantly enhance the question-level accuracy. As the model is encouraged to explore during the iteration phase, it undertakes broader and bolder attempts—often combining additions across more digits in skipped steps. With the integration of these data, the model trained on this expanded iterative dataset also shows a more pronounced ability to solve OOD problems. As seen in Figure 5(b), the model increasingly employs multi-digit additions in single-step operations. Furthermore, as illustrated in Figure 5(c), there is an improvement in the accuracy of these skipped single-step operations. We believe this may be due to the model-generated data during self-iterations, which are more conducive to enhancing its capability to skip steps, thereby benefiting from this process.

### 5.4 Accuracy of Step-Skipping Answers

Figure 6 shows the step skipping behavior and accuracy of the standard models at each iteration on the Analog of Algebra task using Llama-2. The Skipping Ratio measures how often the model skips steps in the test set, while Accuracy reflects the correctness of these skipping answers.

We observe that in the beginning models inherently struggle in OOD scenarios, often producing reasoning steps that are incomplete or shorter than the problem complexity requires. In "cold start" settings, where the model is trained solely with complete steps, it performs well with in-domain questions but fails to maintain complete reasoning steps and tends to generate shorter responses on OOD sets. Due to its limited generalizability, these skipping or missing steps negatively impact the performance. However, as the model progressively adapts to step skipping over iterations, the accuracy of the shorter responses improves, suggesting it gradually develops a more reliable ability to skip steps when appropriate. Analysis across all tasks can be found in Appendix B.4.

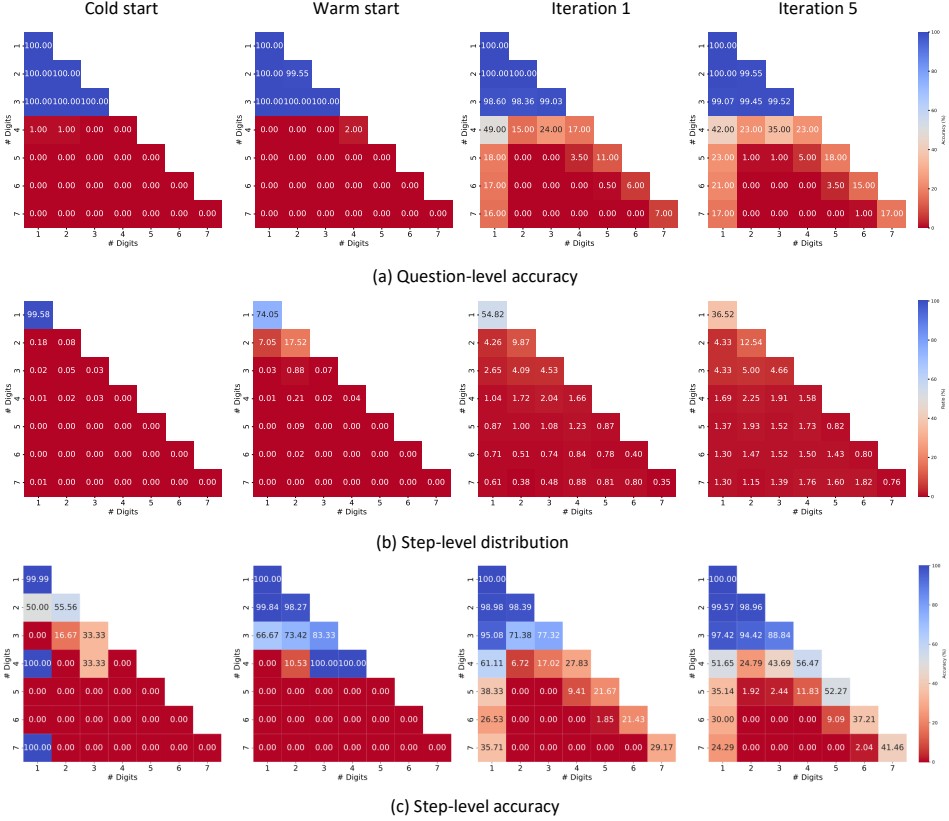

Figure 5: Model behavior analysis on the test set of multi-digit addition task. Initially constrained to single-digit additions, the model progressively incorporates multi-digit calculations with skipped steps through iterative learning, showing an enhancement in solving out-of-distribution problems and executing more complex calculations with higher accuracy.

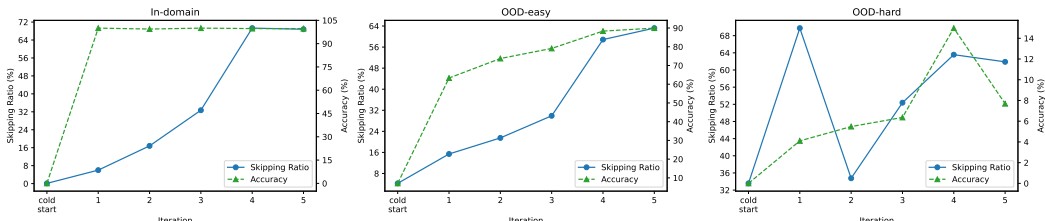

Figure 6: Skipping ratio and the accuracy of the skipping responses on Analog of Algebra.

## 5.5 Analysis on the Influence of Training Steps

Throughout the iterations, as the model progressively generates more successful step skipping data, the size and the quality of the resulting dataset also gradually increases. This can be considered as a special form of augmentation for answer diversity. To investigate whether the performance improvements are primarily due to the model learning from more training steps, we increase the number of training epochs during the initialization phase to match the data volume after iterations. The comparison results shown in Table 4 reveal that increasing the number of training epochs does not always lead to performance enhancements; instead, it may cause a performance decline due to overfitting. In contrast, mixing skip-step data from the iterative process not only maintains or improves performance in in-domain and OOD-easy tasks but also achieves consistent gains in OOD-hard setting. When the total number of training steps is similar, the integration of skipping data yields better performance.

Table 4: Performance comparison across different tasks with varying training steps.

| Task | Iteration | # steps | In-domain | OOD-easy | OOD-hard |
|------|-----------|---------|-----------|----------|----------|
| Analog of Algebra | Cold start - ep4 | 2.9K | 99.9 | 89.7 | 4.5 |
| | Cold start - ep5 | 3.6K | 100 | 84.9 | 2.4 |
| | Iteration 5 - ep2 | 3.3K | 99.8 | 90.5 | 14.3 |
| Multi-digit Addition | Cold start - ep5 | 1.8K | 100 | 0 | 0 |
| | Cold start - ep6 | 2.2K | 100 | 0 | 0 |
| | Warm start - ep2 | 1.4K | 99.9 | 0 | 0.1 |
| | Warm start - ep3 | 2.1K | 100 | 0.1 | 0 |
| | Iteration 5 - ep2 | 2.0K | 99.5 | 13.5 | 5.8 |
| Directional Reasoning | Cold start - ep3 | 0.8K | 100 | 91.2 | 43.2 |
| | Cold start - ep4 | 1.0K | 100 | 91.0 | 34.8 |
| | Warm start - ep2 | 1.0K | 100 | 90.6 | 43.4 |
| | Warm start - ep3 | 1.5K | 100 | 84.6 | 34.4 |
| | Iteration 5 - ep2 | 1.0K | 100 | 90.4 | 56.2 |

## 5.6 Extended Iterative Training

In this section, we extend the iterative process to allow the model to skip up to 4 steps, rather than restricting it to less than 2 steps on Analog of Algebra. The process is continued for a total of 9 iterations, and the results are shown in Figure 7. The model continues to benefit from additional iterations beyond Iteration 5, which serves as the default cutoff in our main results. Specifically, the accuracy on the OOD-hard set improves steadily, reaching over 18% by the ninth iteration. This increase suggests that even with a greater allowance for step-skipping, the model's ability to generalize to harder out-of-domain samples is enhanced with continued training.

Simultaneously, the average number of steps taken decreases across all test sets as iterations progress, suggesting that the model is converging towards fewer steps and becoming increasingly efficient. By the ninth iteration, the step count appears to plateau, indicating that the model has likely reached a stable balance between accuracy and efficiency. We hope our work provides a fresh perspective on exploring the connection between System 2 slow reasoning and System 1 fast thinking, and on facilitating their transformation, paving the way for future research in this direction.

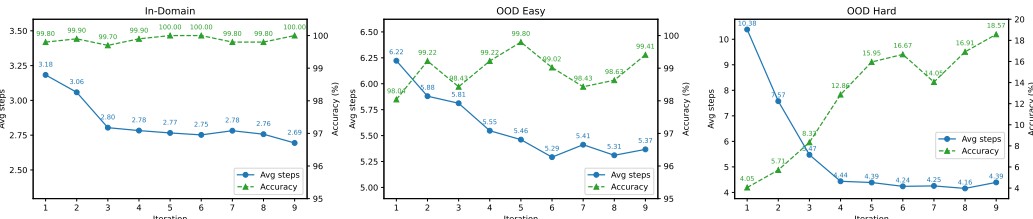

Figure 7: Performance of phi-3-mini across 9 iterations with relaxed step-skipping constraints (up to 4 steps) on Analog of Algebra. The figures show the changes in average steps taken (left y-axis) and accuracy (right y-axis). Continuous iteration improves OOD-hard accuracy and reduces the average number of steps, converging towards stability.

## 6 Conclusion

In this work, we explore the human-like ability of step skipping in language models, providing initial empirical evidence that models can skip steps and benefit from such cognitive behaviors. Addressing the absence of intrinsic motivation for step skipping in models, we design an approach that not only enables models to spontaneously develop the ability but also iteratively encourages models to actively adopt and enhance this behavior. Through experiments on three tasks, we demonstrate that models equipped with step-skipping capabilities can solve tasks more efficiently in fewer steps, without sacrificing accuracy. Further empirical results suggest that training on easy data containing both full steps and skipped reasoning steps can potentially help models generalize to harder scenarios. We hope this work offers insights into the relationship and transition between System 1 and System 2 thinking and contributes to advancing easy-to-hard generalization in language model reasoning.

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

# A Limitations

Our work serves as a preliminary exploration of human-like step skipping capabilities in models, focusing solely on the expansion of problem types in terms of length and compositional complexity, without extending to advanced problem difficulty generalization. We also recognize that ideally there should be a clear criterion for determining when to terminate iterations. We observe that the model can also perform better in intermediate rounds, which suggests the need for further adjustment of this hyperparameter. Additionally, for the convenience in evaluation, our investigations were confined to three simple yet representative tasks. While our designed method can be applied to practical tasks, we leave the exploration of scalability to complex reasoning scenarios as future work.

# B Appendix

## B.1 Details of data creation

### B.1.1 Training data creation

For the Analog of Algebra task, we ensure the quality of the auto-generated dataset by creating full-step reasoning data using standard algebraic rules applied to operators. To further verify the validity and consistency of the intermediate steps, we utilize the SymPy [33] library. Specifically, we perform SymPy simplification for each intermediate step and ensure that the resulting equation remains algebraically equivalent to the final simplified answer.

For the Multi-digit Addition task, the internal results are generated using Python's built-in calculation modules, ensuring accurate computations.

For the Directional Reasoning task, the clarity of the question formulation guarantees that all intermediate steps are 100% correct. Each step is derived through rule-based decomposition, ensuring the correctness of the intermediate steps.

### B.1.2 Manual skipping data for warm start

We define several heuristic rules to create skipping data for warm start initialization. For the multi-digit addition task, we randomly merge two single-digit addition steps to form a two-digit addition step. For the directional reasoning task, we incorporate more human expertise by skipping steps that involve two adjacent directions that result in no change. For example, adjacent actions such as "right-left", "left-right", and "around-around" will not alter the final facing direction, so we manually skip these steps. We only manually create one skipped step within a single data.

## B.2 Skipping data accuracy trend in cold start

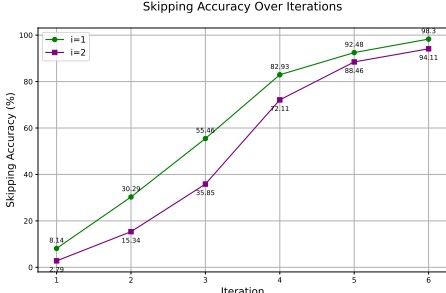

Figure 8: Skipping data accuracy change during cold start in Analog of Algebra.

From Figure 8, the number of correct skipping data keeps increasing as the iterations progress. Higher accuracy results in more valid data involved in the mixed dataset. This iterative approach allows the model to gradually develop the step skipping ability and produce more valid data with fewer steps.

## B.3 Detailed results of each iteration

Table 5 and Table 6 show the detailed performance of standard finetuned models from each iteration on Llama2-7B and phi-3-mini respectively. We report the average performance and the standard deviation across three runs with different random seeds.

Analyzing the results from each iteration, we find that the final iteration does not consistently yield the best performance, highlighting the importance of identifying an optimal stopping point as a direction for future work. Additionally, significant fluctuations are observed in the test results, particularly in the OOD settings. Therefore, developing a more stable approach for OOD generalization tasks is another potential area for further exploration.

Table 5: Performance comparison of models from different iterations on Llama-7B. "Avg steps" denotes the average number of steps taken in the prediction.

| Task | Iteration | In-domain | | OOD-easy | | OOD-hard | |
|------|-----------|-----------|----------|----------|----------|----------|----------|
| | | Acc | Avg steps | Acc | Avg steps | Acc | Avg steps |
| Analog of Algebra | Cold start | $99.87_{0.12}$ | $3.19_{0.00}$ | $85.91_{1.65}$ | $4.79_{0.04}$ | $7.94_{4.91}$ | $11.57_{1.37}$ |
| | Iter 1 | $99.77_{0.15}$ | $3.13_{0.01}$ | $86.72_{1.60}$ | $4.65_{0.06}$ | $8.65_{3.81}$ | $11.05_{1.31}$ |
| | Iter 2 | $99.77_{0.06}$ | $3.04_{0.02}$ | $88.93_{2.16}$ | $4.69_{0.11}$ | $5.88_{1.58}$ | $16.44_{1.29}$ |
| | Iter 3 | $99.90_{0.17}$ | $2.89_{0.05}$ | $88.47_{1.22}$ | $4.50_{0.09}$ | $6.03_{2.78}$ | $12.32_{2.38}$ |
| | Iter 4 | $99.93_{0.12}$ | $2.53_{0.07}$ | $\mathbf{90.77_{0.30}}$ | $4.19_{0.12}$ | $\mathbf{8.57_{5.15}}$ | $11.39_{1.47}$ |
| | Iter 5 | $99.80_{0.10}$ | $2.43_{0.13}$ | $90.67_{1.88}$ | $4.05_{0.17}$ | $8.10_{1.26}$ | $10.92_{0.89}$ |
| Multi-digit Addition | Cold start | $100.0_{0.00}$ | $2.86_{0.00}$ | $0.06_{0.10}$ | $3.25_{0.04}$ | $0.00_{0.00}$ | $3.69_{0.06}$ |
| | Warm start | $99.53_{0.32}$ | $2.72_{0.24}$ | $0.14_{0.13}$ | $3.02_{0.38}$ | $0.11_{0.10}$ | $3.49_{0.37}$ |
| | Iter 1 | $99.07_{0.23}$ | $1.75_{0.11}$ | $14.36_{2.75}$ | $1.85_{0.08}$ | $4.06_{0.89}$ | $2.18_{0.18}$ |
| | Iter 2 | $98.87_{0.12}$ | $1.45_{0.07}$ | $14.11_{1.54}$ | $1.54_{0.07}$ | $4.44_{1.84}$ | $2.05_{0.08}$ |
| | Iter 3 | $99.13_{0.23}$ | $1.46_{0.04}$ | $\mathbf{16.81_{1.70}}$ | $1.53_{0.08}$ | $4.06_{1.00}$ | $2.00_{0.13}$ |
| | Iter 4 | $98.77_{0.06}$ | $1.41_{0.04}$ | $16.08_{4.01}$ | $1.49_{0.05}$ | $\mathbf{5.13_{1.17}}$ | $2.08_{0.11}$ |
| | Iter 5 | $99.17_{0.35}$ | $1.46_{0.04}$ | $13.97_{0.42}$ | $1.49_{0.20}$ | $4.75_{0.87}$ | $2.06_{0.26}$ |
| Directional Reasoning | Cold start | $100.0_{0.00}$ | $7.01_{0.00}$ | $\mathbf{90.00_{0.53}}$ | $15.77_{0.46}$ | $42.00_{6.24}$ | $19.39_{0.29}$ |
| | Warm start | $99.97_{0.06}$ | $6.28_{0.04}$ | $87.20_{5.21}$ | $14.65_{0.43}$ | $42.33_{9.25}$ | $18.02_{2.4}$ |
| | Iter 1 | $100.0_{0.00}$ | $6.46_{0.04}$ | $83.00_{5.57}$ | $14.69_{0.14}$ | $29.47_{6.59}$ | $14.24_{2.86}$ |
| | Iter 2 | $99.97_{0.06}$ | $6.44_{0.06}$ | $86.47_{3.93}$ | $14.95_{0.84}$ | $40.67_{13.30}$ | $17.42_{3.23}$ |
| | Iter 3 | $100.0_{0.00}$ | $6.49_{0.13}$ | $88.60_{1.64}$ | $14.93_{0.44}$ | $41.53_{7.30}$ | $17.60_{0.68}$ |
| | Iter 4 | $99.90_{0.10}$ | $6.36_{0.06}$ | $89.20_{2.03}$ | $14.66_{0.30}$ | $44.33_{6.99}$ | $17.79_{1.31}$ |
| | Iter 5 | $100.0_{0.00}$ | $6.45_{0.06}$ | $89.33_{1.36}$ | $14.87_{0.12}$ | $\mathbf{51.80_{4.21}}$ | $19.49_{0.79}$ |

**Cold start vs. warm start**   In the Multi-digit Addition task, we observe that phi-3-mini achieves satisfactory results with cold start training alone, allowing the model to enter the iteration phase without relying on manually provided skipping data. Table 7 shows the model's performance when initialized with a cold start in Multi-digit Addition. Compared to the results in Table 6, where the model begins with a warm start, the cold start approach enables the model to independently explore and develop its skipping behaviors. This leads to a more pronounced improvement in the OOD settings, with accuracy of 25.06% versus 14.98% in Iteration 5 on OOD-hard. Additionally, we observe that while warm start enables a more immediate reduction in steps, cold start shows a more gradual decrease in the number of steps taken.

## B.4 Accuracy of step-skipping answers

In this section, we provide the ratio and the accuracy of the skipping responses across three tasks using both base models. The results are shown in Figure 9 and Figure 10. In general, the models demonstrate a progressively enhanced step skipping capability across various test settings for all tasks. In most cases, the model increasingly favors adopting more skipped reasoning steps over iterations, with the accuracy of skipped responses also improving correspondingly. However, we observe that the proportion of skipped responses fluctuates across different stages of iteration, rather than following a strictly monotonic trend. Given that the model autonomously decides whether to employ skipping, this pattern may indicate the model's attempt to find a balance between using step

Table 6: Performance comparison of models from different iterations on phi-3-mini. "Avg steps" denotes the average number of steps taken in the prediction.

| Task | Iteration | In-domain | | OOD-easy | | OOD-hard | |
|---|---|---|---|---|---|---|---|
| | | Acc | Avg steps | Acc | Avg steps | Acc | Avg steps |
| Analog of Algebra | Cold start | $99.60_{0.10}$ | $3.19_{0.01}$ | $98.04_{1.09}$ | $6.16_{0.00}$ | $4.05_{2.11}$ | $10.01_{0.32}$ |
| | Iter 1 | $99.77_{0.06}$ | $3.18_{0.00}$ | $99.02_{0.34}$ | $6.14_{0.02}$ | $3.17_{3.64}$ | $9.82_{0.69}$ |
| | Iter 2 | $99.83_{0.12}$ | $3.13_{0.02}$ | $98.89_{1.08}$ | $6.07_{0.01}$ | $5.40_{2.74}$ | $9.00_{0.36}$ |
| | Iter 3 | $99.90_{0.10}$ | $2.95_{0.05}$ | $99.54_{0.11}$ | $5.89_{0.09}$ | $9.92_{3.47}$ | $7.67_{0.39}$ |
| | Iter 4 | $99.97_{0.06}$ | $2.71_{0.06}$ | $99.41_{0.00}$ | $5.62_{0.22}$ | $10.16_{0.96}$ | $7.34_{0.11}$ |
| | Iter 5 | $99.90_{0.17}$ | $2.75_{0.28}$ | $98.95_{0.23}$ | $5.60_{0.33}$ | $11.13_{1.50}$ | $7.98_{0.44}$ |
| Multi-digit Addition | Cold start | $99.92_{0.13}$ | $2.86_{0.00}$ | $35.93_{12.29}$ | $5.03_{0.22}$ | $5.39_{1.90}$ | $5.44_{0.17}$ |
| | Warm start | $99.97_{0.06}$ | $2.62_{0.07}$ | $39.08_{3.87}$ | $3.80_{0.35}$ | $5.11_{2.62}$ | $4.06_{0.44}$ |
| | Iter 1 | $99.87_{0.15}$ | $2.21_{0.06}$ | $45.03_{6.98}$ | $2.43_{0.30}$ | $12.36_{0.66}$ | $2.55_{0.34}$ |
| | Iter 2 | $99.93_{0.06}$ | $2.02_{0.13}$ | $49.45_{5.18}$ | $2.22_{0.15}$ | $13.88_{3.84}$ | $2.42_{0.06}$ |
| | Iter 3 | $99.93_{0.12}$ | $2.13_{0.08}$ | $43.08_{5.80}$ | $2.30_{0.13}$ | $13.54_{1.39}$ | $2.57_{0.07}$ |
| | Iter 4 | $99.87_{0.15}$ | $2.01_{0.05}$ | $45.25_{9.93}$ | $2.28_{0.11}$ | $12.84_{1.10}$ | $2.52_{0.24}$ |
| | Iter 5 | $99.93_{0.06}$ | $2.08_{0.12}$ | $46.61_{12.70}$ | $2.31_{0.11}$ | $14.98_{3.19}$ | $2.59_{0.12}$ |
| Directional Reasoning | Cold start | $99.83_{0.36}$ | $7.01_{0.00}$ | $91.47_{3.68}$ | $15.46_{0.25}$ | $62.67_{18.21}$ | $24.85_{0.43}$ |
| | Warm start | $99.80_{0.17}$ | $6.82_{0.17}$ | $93.67_{1.94}$ | $15.19_{0.07}$ | $71.80_{5.30}$ | $24.61_{0.15}$ |
| | Iter 1 | $99.93_{0.12}$ | $6.48_{0.15}$ | $94.40_{1.51}$ | $14.94_{0.13}$ | $73.13_{6.93}$ | $24.43_{0.30}$ |
| | Iter 2 | $99.97_{0.06}$ | $6.36_{0.10}$ | $95.33_{2.42}$ | $14.72_{0.11}$ | $74.80_{8.67}$ | $24.26_{0.63}$ |
| | Iter 3 | $99.67_{0.35}$ | $6.40_{0.13}$ | $94.47_{1.70}$ | $14.83_{0.13}$ | $75.40_{6.39}$ | $24.24_{0.60}$ |
| | Iter 4 | $99.60_{0.35}$ | $6.23_{0.12}$ | $95.13_{0.95}$ | $14.72_{0.29}$ | $72.87_{11.43}$ | $24.20_{0.59}$ |
| | Iter 5 | $99.70_{0.17}$ | $6.12_{0.06}$ | $93.73_{0.70}$ | $14.44_{0.04}$ | $73.87_{4.17}$ | $23.77_{0.18}$ |

Table 7: Performance across iterations in the Multi-digit Addition task with the phi-3-mini model, initialized from a cold start rather than a warm start.

| Task | Iteration | In-domain | | OOD-easy | | OOD-hard | |
|---|---|---|---|---|---|---|---|
| | | Acc | Avg steps | Acc | Avg steps | Acc | Avg steps |
| Multi-digit Addition | Cold start | $99.92_{0.13}$ | $2.86_{0.00}$ | $35.93_{12.29}$ | $5.03_{0.22}$ | $5.39_{1.90}$ | $5.44_{0.17}$ |
| | Warm start | $99.97_{0.06}$ | $2.62_{0.07}$ | $39.08_{3.87}$ | $3.80_{0.35}$ | $5.11_{2.62}$ | $4.06_{0.44}$ |
| | Iter 1 | $100.0_{0.00}$ | $2.83_{0.05}$ | $37.44_{12.73}$ | $5.03_{0.18}$ | $5.21_{0.72}$ | $5.28_{0.17}$ |
| | Iter 2 | $100.0_{0.00}$ | $2.78_{0.15}$ | $38.50_{28.87}$ | $4.77_{0.57}$ | $4.83_{4.05}$ | $5.00_{0.60}$ |
| | Iter 3 | $99.90_{0.10}$ | $2.78_{0.07}$ | $58.78_{9.73}$ | $5.03_{0.20}$ | $9.04_{0.66}$ | $5.27_{0.13}$ |
| | Iter 4 | $99.93_{0.06}$ | $2.38_{0.28}$ | $49.19_{16.52}$ | $4.18_{0.78}$ | $25.35_{11.73}$ | $4.95_{0.27}$ |
| | Iter 5 | $99.83_{0.15}$ | $2.54_{0.27}$ | $55.47_{3.49}$ | $4.51_{0.32}$ | $25.06_{6.79}$ | $5.29_{0.13}$ |

skipping and providing full-step solutions. Exclusively relying on skipping would not necessarily be the optimal answering strategy. We also find that a warm start significantly boosts the model's skipping behavior. Consequently, in models with a warm start, the changes across iterations are less pronounced, though overall accuracy still improves.

## B.5 Data mixing choices for standard model training

Table 8: Ablation of different data mixing choices on Analog of Algebra.

| Training data | In-domain | | OOD-easy | | OOD-hard | |
|---|---|---|---|---|---|---|
| | Acc | Avg steps | Acc | Avg steps | Acc | Avg steps |
| Skipping | 98.70 | 1.94 | 93.66 | 4.97 | 7.86 | 7.44 |
| Skipping w/ Cold start | **99.90** | 2.75 | **98.95** | 5.60 | **11.13** | 7.98 |

In this section, we analyze the role of data mixture in iterative training and its effect on the performance of standard models $M^{\texttt{standard}}$. Specifically, we examine how the inclusion of both cold-start data and generated skipping data enhances the model's generalization ability and comprehension of complex

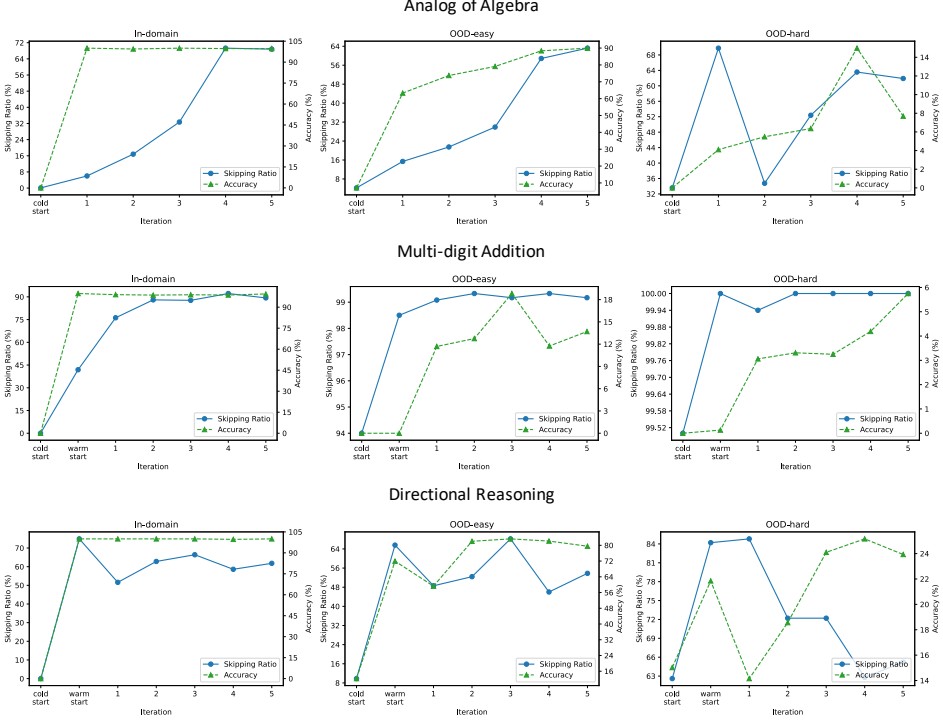

Figure 9: Skipping ratio and accuracy at each iteration on Llama2-7B.

reasoning paths. Table 8 presents an ablation study comparing different data mixing strategies with phi-3-mini model on the Analog of Algebra task. The "Skipping" setting utilizes only the generated skipping data $D'_{k-1}$ for training the standard model $M_k$, while "w/ Cold Start" incorporates both the original cold-start data and the skipping data, which serves as the default configuration in our experiments. The analysis is based on data from Iteration 5, and we report average performance across three runs with different random seeds. Our findings suggest that relying solely on skipping data may limit the model's capacity to address OOD scenarios. Although skipping data provides shorter average steps, it lacks the complete reasoning steps essential for a comprehensive understanding of the task, potentially leading the model to depend on shortcuts that harm generalization. By incorporating a mixture of cold-start and skipping data, the model is able to learn from both complete and skipped reasoning chains, which enables a more robust understanding, supporting stronger generalization capabilities.

## B.6 Cross-Domain Generalization of Step-Skipping Ability

Table 9: Cross-domain generalization of step-skipping capability in the phi-3-mini model. In the specified "Withheld Task" setting, step-skipping data is excluded from one specific task, while the "All" setting includes only full-step data across three tasks.

| Evaluation Task | Withheld Task | In-domain | | OOD-easy | | OOD-hard | |
| --- | --- | --- | --- | --- | --- | --- | --- |
| | | Acc | Avg steps | Acc | Avg steps | Acc | Avg steps |
| Analog of Algebra | All | 51.3 | 2.65 | 44.5 | 5.58 | 1.9 | 10.68 |
| | Analog of Algebra | **53.9** | 2.71 | **56.9** | 5.74 | **7.1** | 10.97 |
| Multi-digit Addition | All | 100.0 | 2.86 | 22.4 | 4.71 | 4.2 | 5.39 |
| | Multi-digit Addition | 95.7 | 2.59 | **34.3** | 4.75 | 2.4 | 5.35 |
| Directional Reasoning | All | 100.0 | 7.01 | 96.0 | 15.46 | 75.8 | 25.03 |
| | Directional Reasoning | 97.8 | 6.98 | **96.2** | 15.42 | **80.0** | 24.92 |

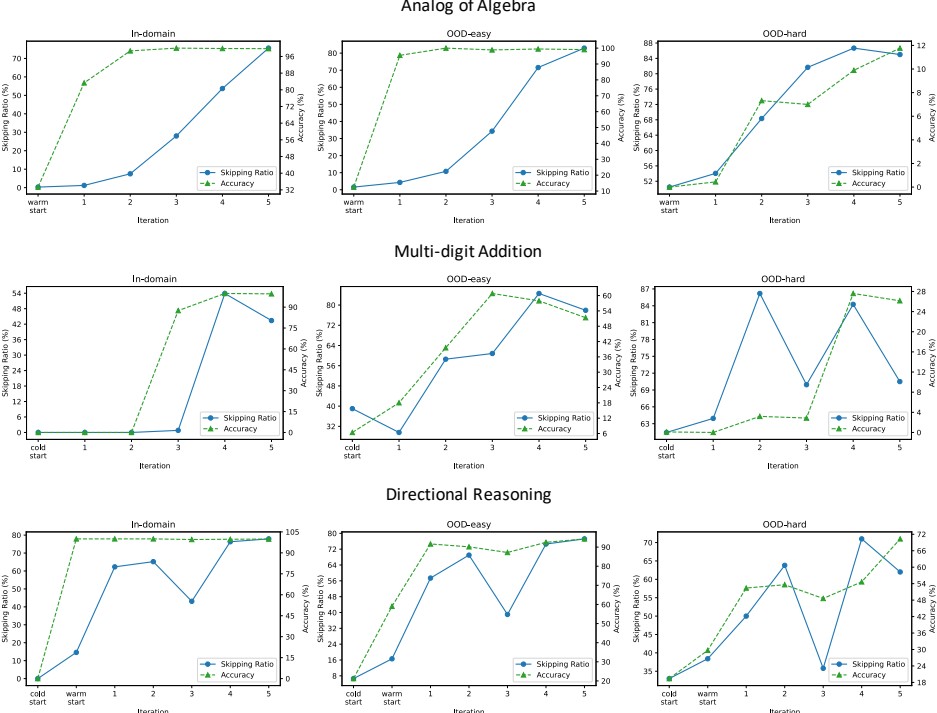

Figure 10: Skipping ratio and accuracy at each iteration on phi-3-mini. On Multi-digit Addition, we illustrate the analysis of the model that is initialized from Cold start.

To investigate the cross-domain generalization of step-skipping capabilities, we conduct a controlled experiment to assess the impact of step-skipping training data from one task on the model's performance in others. Specifically, we sampled 2,000 training examples per dataset, including 1,600 step-skipping answers in which exactly one step was successfully skipped from these samples, all from Iteration 5. This setup ensures an equal balance of full-step and step-skipping data across all three tasks.

We use the phi-3-mini model across three tasks, with the "withheld task" representing the task that lacks step-skipping data during training. The "All" setting contains only full-step answers for all tasks, with no step-skipping data included. The configurations are as follows:

- All setting: task1-full + task2-full + task3-full

- Withheld setting: task1-full + task1-skip + task2-full + task2-skip + task3-full

Table 9 summarizes the model's performance on each evaluation task. The withheld task's results are compared to those from the "All" setting, where all tasks are trained with only full-step answers. Our findings reveal that step-skipping data in one or more tasks positively affects the performance of the withheld task. In most cases, models trained with step-skipping data from other tasks exhibit improved accuracy and step-skipping performance across datasets, maintaining a comparable number of steps to the "All" setting. For example, in the Analog of Algebra task, the average steps remain similar, yet accuracy improvements are observed in OOD settings, indicating that training with step-skipping data promotes a transferable ability to reason efficiently across domains. The overall accuracy increase suggests that inclusion of step-skipping data in some tasks enables the model to generalize this ability, even when explicit step-skipping examples are unavailable in the target task. These results suggest that the step-skipping capability learned in one domain can generalize across different tasks, underscoring the potential for enhancing model efficiency through strategic data composition.

## B.7 Experiments on GSM8K

In addition to the synthetic datasets analyzed in the main body of the paper, we conduct experiments on GSM8K [8] to evaluate the applicability of our method to more complicated tasks. To create a controlled experimental setting, we classify data requiring no more than 4 steps in the annotated answers as in-domain data and the remaining as out-of-domain data. Table 10 provides an overview of the dataset splits.

Table 10: Dataset split for GSM8K.

| Splits | In-domain | Out-of-domain | Total |
|---|---|---|---|
| Train | 6,235 | 1,238 | 7,473 |
| Test | 1,094 | 225 | 1,319 |

The results across different iterations is presented in Table 11. We observe that while the average number of reasoning steps per iteration progressively declines, the accuracy remains stable across iterations. Several factors may explain the limited improvement in accuracy. Analysis of the model's step-skipping behavior reveals that intermediate steps frequently contain errors, indicating limitations in the model's ability for effective step reduction. Throughout the iterations, the model struggles to generate responses in fewer steps, as the complexity of the questions often necessitates a complete reasoning chain to reach a solution. This aligns with findings by Yu et al. [49], which suggest that CoT reasoning is difficult to distill into System 1. We consider further exploration of the gradual transition between System 1 and System 2 thinking, particularly for complex tasks, as a promising direction for future research.

Table 11: Performance comparison across different iterations. The table shows accuracy and average steps for various test and training datasets.

| Iteration | Test-ID | | Test-OOD | | Train-OOD | |
|---|---|---|---|---|---|---|
| | Acc | Steps | Acc | Steps | Acc | Steps |
| Cold start | 79.89 | 4.23 | 61.33 | 6.5 | 63.33 | 5.99 |
| Iter1 | 78.06 | 4.24 | 59.56 | 5.9 | 64.62 | 5.96 |
| Iter2 | 78.52 | 4.15 | 57.78 | 5.84 | 63.33 | 6.02 |
| Iter3 | 79.16 | 4.19 | 52.44 | 5.86 | 63.57 | 5.90 |
| Iter4 | 75.69 | 4.16 | 56.44 | 5.78 | 63.97 | 5.88 |
| Iter5 | 78.43 | 4.08 | 60.44 | 5.77 | 61.55 | 5.72 |

## B.8 Case study

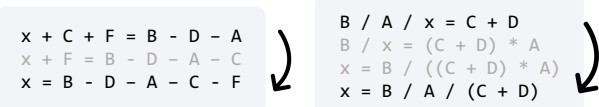

Figure 11: Frequent skipping pattern in Analog of Algebra (translated into standard algebra).

Figure 11 shows two frequent cases where models spontaneously develop to skip certain steps in iterations. Note that in Analog of Algebra, we employ the cold start setting in initialization. The behaviors shown in the figure emerge solely from the model itself, which indicates the model has developed its own step skipping preference. In addition, we show two cases in Figure 12 from Analog of Algebra and Multi-digit Addition tasks. In these examples, the full step answers exhibit errors in their reasoning processes. In contrast, the skipped step answers choose to skip steps and arrive at the correct answer in fewer steps.

```
Analog of Algebra

Q: D / G + I + B = A + C * H * F / x

Full-step answer:
A + C * H * F / x = D / G + I + B
C * H * F / x = D / G + I + B - A
H * F / x = D / G / C + I / C + B - A / C
F / x = D / G / C / H + I / C / H + B - A / C / H
x = F / ( D / G / C / H + I / C / H + B - A / C / H )

Skipped-step answer:
A + C * H * F / x = D / G + I + B
C * H * F / x = D / G + I + B - A
C * F / x = D / G / H + I / H + B / H - A / H
x = C * F / ( D / G / H + I / H + B / H - A / H )

Multi-digit Addition

Q: 3909 + 641

Full-step answer:
1. 0 + 1 (plus carry 0) = 1 (write down 1, no carry over).
2. 9 + 4 (plus carry 0) = 13 (write down 3, carry over 1).
3. 3 + 6 (plus carry 1) = 10 (write down 0, carry over 1).
4. Carry the final carry over 1 to the next place.
Thus, the answer is 1031.

Skipped-step answer:
1. 9 + 1 (plus carry 0) = 10 (write down 0, carry over 1).
2. 90 + 64 (plus carry 1) = 155 (write down 55, carry over 1).
3. 3 + 0 (plus carry 1) = 4 (write down 4, no carry over).
Thus, the answer is 4550.
```

Figure 12: Case study of skipped-step answers in Analog of Algebra and Multi-digit Addition.

