# OpenReview forum: "Can Language Models Learn to Skip Steps?"
_NeurIPS.cc/2024/Conference — NeurIPS 2024 poster_

### Official Review · Reviewer_E6Qj · 2024-06-12

**Soundness:** 2
**Presentation:** 3
**Contribution:** 2
**Rating:** 5
**Confidence:** 4

**Summary:**

The paper explores the ability of language models to skip steps in their reasoning processes. The authors introduce a controlled framework to stimulate step-skipping behavior by iteratively refining models to generate shorter and accurate reasoning paths. The study demonstrates that models can develop this ability under guidance, leading to increased efficiency without sacrificing accuracy. The paper presents empirical results showing enhanced generalization capabilities in out-of-domain scenarios after fine-tuning on expanded datasets that include both complete and skipped reasoning sequences.

**Strengths:**

- The empirical results are robust in three domains, showing benefits in efficiency of the proposed method.
- The paper is clearly written and well-organized, making it easy to follow the authors' methodology and findings.

**Weaknesses:**

- Only one backbone model is considered. Experiments across model families and model sizes should be considered to show the generalization ability of the proposed methods.
- The OOD test is actually the harder in-domain test. I am curious about the across domain effect of the proposed method For example, how's the effect of training on "Analog of Algebra" and test on  "Multi-digit Addition", given that the skip ability should be a general ability across different domains?
- In methodology: "We begin with a training dataset D0, which contains detailed full-step reasoning answers to the questions." -> How the full-step reasoning data created?

**Questions:**

see weakness

**Limitations:**

yes

---

> ### Author Rebuttal · Authors · 2024-08-06
>
> Thank you for your valuable and constructive feedback. We provide specific responses and clarifications as follows.
>
>
> **[W1]**
>
> Thank you for the suggestion. We acknowledge the importance of testing our method across different models. Following your advice, we performed additional experiments on Phi-3-mini for the Analog of Algebra task. These experiments confirm that our method is generalizable to other model series. We will include the results across all tasks in the revision.
>
>
> |          | In-domain    |            | OOD-easy   |            | OOD-hard   |            |
> |----------|--------------|------------|------------|------------|------------|------------|
> |          | Acc          | Avg steps  | Acc        | Avg steps  | Acc        | Avg steps  |
> | Cold Start | 99.5       | 3.18       | 96.86      | 6.16       | 1.67       | 9.83       |
> | Iter 1   | 99.7         | 3.18       | 99.22      | 6.13       | 0.95       | 10.61      |
> | Iter 2   | 99.9         | 3.11       | 100        | 6.06       | 5.48       | 9.06       |
> | Iter 3   | 99.9         | 2.9        | 99.6       | 5.79       | 6.67       | 7.74       |
> | Iter 4   | 99.9         | 2.64       | 99.41      | 5.36       | 9.05       | 7.46       |
> | Iter 5   | 99.7         | 2.43       | 98.82      | 5.23       | 10.24      | 7.77       |
>
> *Table: Phi-3-mini on Analog of Algebra.*
>
> **[W2]**
>
> Thank you for your valuable suggestion! We acknowledge the importance of evaluating the generalizability of our method. We are actively working on additional experiments and will provide an update on the results once the experiments are completed.
>
>
> **[W3]**
>
> For the three datasets used in our work, we use heuristic rules to automatically generate the reasoning processes. For example, in the Analog of Algebra dataset, we first create full-step reasoning data using standard algebra rules on operators. Then, we replace the variables with analog symbols. We also plan to release the code for creating the dataset in the future.

---

> > ### Comment · Reviewer_E6Qj · 2024-08-13
> >
> > Thank you for the responses, given the authors' response, the W2 has not been resolved.  In addition, I am curious about how the authors ensure the quality of the auto-generated dataset. Is there any human evaluation of that?

---

> > > ### Author Response · Authors · 2024-08-14
> > >
> > > Thank you for your valuable feedback! We address the remaining concerns as follows.
> > >
> > > ### **[W2] Generalization across different domains**
> > >
> > > We appreciate the insightful questions regarding the generalization of step-skipping ability across different tasks. To address these concerns, we conduct a controlled experiment. Specifically, we sample 2000 training examples from each dataset, along with 1600 step-skipping answers where exactly one step is successfully skipped, drawn from these 2000 questions in Iteration 5. This approach ensures that all three tasks have an equal amount of full-step and step-skipping data.
> > >
> > > We focus on evaluating a model that requires the number of steps as input. During testing, we instruct the model to solve each problem in $n−1$ steps, where $n$ is the full step count for that problem. This setting allows us to directly assess the step-skipping behavior by measuring accuracy under this evaluation setting. If the accuracy increases under this setting, it indicates improved step-skipping ability.
> > >
> > > We utilize the Phi-3-mini model across all tasks. In the table below, the "withheld task" refers to the task that does not have step-skipping data in the training phase. The "All" setting includes only full-step answers across all three tasks without any step-skipping data.
> > >
> > > To clarify the settings:
> > > * All setting = task1-full + task2-full + task3-full
> > > * Withheld setting = task1-full + *task1-skip* + task2-full + *task2-skip* + task3-full
> > >
> > > *Table 1: Evaluation task - Analog of Algebra*
> > >
> > > | Withheld task       | In-domain | Avg steps | OOD-easy | Avg steps | OOD-hard | Avg steps |
> > > |---------------------|-----------|-----------|----------|-----------|----------|-----------|
> > > | All                 | 51.3      | 2.65      | 44.5     | 5.58      | 1.9      | 10.68     |
> > > | Analog of Algebra   | 53.9      | 2.71      | 56.9     | 5.74      | 7.1      | 10.97     |
> > >
> > > *Table 2: Evaluation task - Multi-digit Addition*
> > >
> > > | Withheld task       | In-domain | Avg steps | OOD-easy | Avg steps | OOD-hard | Avg steps |
> > > |---------------------|-----------|-----------|----------|-----------|----------|-----------|
> > > | All                 | 100.0     | 2.86      | 22.4     | 4.71      | 4.2      | 5.39      |
> > > | Multi-digit Addition| 95.7      | 2.59      | 34.3     | 4.75      | 2.4      | 5.35      |
> > >
> > > *Table 3: Evaluation task - Directional Reasoning*
> > >
> > > | Withheld task          | In-domain | Avg steps | OOD-easy | Avg steps | OOD-hard | Avg steps |
> > > |------------------------|-----------|-----------|----------|-----------|----------|-----------|
> > > | All                    | 100.0     | 7.01      | 96.0     | 15.46     | 75.8     | 25.03     |
> > > | Directional Reasoning  | 97.8      | 6.98      | 96.2     | 15.42     | 80.0     | 24.92     |
> > >
> > > The results show that the presence of step-skipping data in the training of other tasks positively impacts the performance of the withheld task. Compared to the "All" setting, models trained with step-skipping data in other tasks demonstrate improved step-skipping performance across all datasets with a comparable number of steps. Since the model is explicitly instructed to solve problems with fewer steps, the average steps in the withheld task remain similar to the "All" setting. However, the observed boost in accuracy suggests that the model benefits from the generalized step-skipping ability acquired from other tasks. This confirms the generalizability of the step-skipping ability across different domains.
> > >
> > > We appreciate the reviewers' insightful questions and believe that these findings will further strengthen our work. We will incorporate these findings into our revision.
> > >
> > > ### **[Q] How to ensure the quality of the auto-generated dataset?**
> > >
> > > Thank you for your additional question!
> > >
> > > For the Analog of Algebra task, we ensure the quality of the auto-generated dataset by creating full-step reasoning data using standard algebraic rules applied to operators. To further verify the validity and consistency of the intermediate steps, we utilize the SymPy library. Specifically, we perform SymPy simplification for each intermediate step and ensure that the resulting equation remains algebraically equivalent to the final simplified answer.
> > >
> > > For the Multi-digit Addition task, the internal results are generated using Python's built-in calculation modules, ensuring accurate computations.
> > >
> > > For the Directional Reasoning task, the clarity of the question formulation guarantees that all intermediate steps are 100% correct. Each step is derived through rule-based decomposition, ensuring the correctness of the intermediate steps.
> > >
> > > All the data is guaranteed to be correct due to the heuristic nature of the creation process. We will make sure that the data creation and quality control processes are further clarified in the revision.

---

### Official Review · Reviewer_DBKK · 2024-07-07

**Soundness:** 3
**Presentation:** 3
**Contribution:** 2
**Rating:** 6
**Confidence:** 4

**Summary:**

This paper proposes an iterative training method that helps sequence models learn to skip steps. The method starts from a training set with full-length solutions or mixed with some skipped-length solutions. At each stage a model learns these solutions with the instruction “Solve it in n steps” and is prompted to generate shorter answers. Correct shorter answers are added to the training set. The effect of this approach is tested using LlaMa-7B on three tasks, including algebraic evaluations, multi-digit addition, and a direction inference task.

**Strengths:**

The proposed skip reasoning pipeline is interesting and was evaluated against a diverse set of tasks with different levels of OOD generalization tests.

The authors conducted detailed analyses to understand the effect of the training pipeline, e.g., Figure 5 with the multi-digit addition is very informative.

The overall presentation is very clear and easy to follow.

**Weaknesses:**

My main concern is the generalizability of this method. As shown in the paper, the model largely benefited from the warm start setup that includes some skipped problems in the first training set, and has trouble generalizing to problems requiring more steps. One interesting generalization test would be to train on a mixture of all three tasks, but withhold adding skipped step instances for one task, and see if the model can generalize skipping steps on the withheld task.

The shorter answers generated during the iterative process also don't seem quite “generated by the model itself”, as filtering out correct answers would require oracle knowledge. Is it assumed that correct answers are any exact subset of the full-length solution?

**Questions:**

The accuracy metric measures final answer accuracy, are intermediate steps correct?

I'm unsure if it makes sense to read too much into the average step metric when accuracy is low.

In figure 4, what does the accuracy look like for only problems where the model skipped steps?

What do you think make multi-digit addition and directional inference more difficult than the algebra task? Especially that accuracy and average step for OOD problems are still pretty bad for multi-digit addition even with the warm start and a few iterations in.

What is the change in the ratio between D_init and D_skipped over the iteration process?

What’s the range of i (in n-i) of the added skip-step instances in D_skipped under warm start?

Line 221 has incorrect figure reference.

**Limitations:**

Noted in the paper.

---

> ### Author Rebuttal · Authors · 2024-08-06
>
> Thank you for your valuable and constructive feedback. We provide specific responses and clarifications as follows.
>
>
> **[W1]**
>
> Thank you for your valuable suggestion! We acknowledge the importance of evaluating the generalizability of our method. We are actively working on additional experiments and will provide an update on the results once the experiments are completed.
>
>
> **[W2]**
>
> In the iteration phase, we query the model to generate shorter answers on the training set and filter based on the correctness of the final answers. The data used are the labels required for training any model, so our filtering process does not rely on any additional information beyond the training data.
>
> When generating answers with fewer steps, the model determines which steps to skip or omit. Thus, the skipping behaviors are entirely generated by the model itself, and the filtering serves as guidance on how to strengthen such behaviors.
>
> Correct answers can indeed be subsets of the full-length solution. The models may actively omit certain steps from the full-length answer to generate the shorter output.
>
> **[Q1]**
>
> Thank you for this insightful question. Our accuracy metric does not measure the correctness of intermediate steps. Automatically evaluating the correctness of the internal reasoning process is a common issue and can be challenging [1]. Previous work [2] has shown that training with incorrect reasoning traces can still lead to improvements. From our manual analysis of Analog of Algebra on In-domain test set, we observe 50 skipped answers and find that 96% of them are correct in its reasoning process. We'll provide the analysis across all tasks in the revision.
>
> [1]Solving math word problems with process- and outcome-based feedback. \
> [2]MetaMath: Bootstrap Your Own Mathematical Questions for Large Language Models.
>
>
> **[Q2]**
>
> The primary motivation of this metric is to measure whether the model has learned the ability to skip steps. Even when the accuracy is not high, the model still uses fewer steps overall while maintaining its overall accuracy. This consistent performance, both when accuracy is low and high, indicates that the model is developing this step-skipping ability and demonstrating it on both in-domain and out-of-domain data.
>
> **[Q3]**
>
> Thank you for your insightful question. We will include the analysis across all tasks in the revision. Here we present the results for the Analog of Algebra task. The Skipping Ratio indicates the proportion of answers where steps were skipped in the entire test set. The Acc measures the accuracy of these skipping answers. The results show that as iterations progress, the model tends to skip steps in more problems, and the accuracy increases accordingly. In the OOD-hard setting, due to the lengthy and difficult nature of the problems, the model initially tends to use shorter answers with lower accuracy. However, the overall accuracy increases with each iteration.
>
>
>
> |         | in-domain   |                   | OOD-easy   |                   | OOD-hard    |                   |
> |---------|--------------------|-------------------|-------------------|-------------------|-------------------|-------------------|
> |         | Skipping Ratio          | Acc         | Skipping Ratio           | Acc         | Skipping Ratio           | Acc         |
> | Iter 1  | 6.0                  | 100               | 15.4               | 63.3             | 69.8              | 4.1              |
> | Iter 2  | 16.8               | 99.4              | 21.5               | 73.7             | 34.8              | 5.5              |
> | Iter 3  | 32.7               | 100               | 29.9               | 79.1              | 52.4              | 6.4              |
> | Iter 4  | 69.3               | 99.7              | 58.9              | 88.4             | 63.6              | 15.0             |
> | Iter 5  | 68.8               | 99.7              | 63.2               | 90.0             | 61.9               | 7.7              |
>
> *Table: Skipping ratio across the iterations and the accuracy of problems with skipping-step answers.*
>
> **[Q4]**
>
> These two tasks involve more steps and actions in the answers. From our manual observation, we noticed that the model tends to overfit on the multi-digit addition task. As the training data only contains numbers with up to three digits, the model overfits to three-digit additions. Here is an incorrect answer from the model's prediction:
>
> Q: 4423 + 684\
> Prediction:
> 1. Add the 10^0's place digits: 2 + 4 (plus carry 0) = 6 (write down 6, no carry over).
> 2. Add the 10^1's place digits: 4 + 8 (plus carry 1) = 12 (write down 2, carry over 1).
> 3. Add the 10^2's place digits: 4 + 6 (plus carry 1) = 11 (write down 1, carry over 1).
> 4. Carry the final carry over 1 to the next place. \
> Thus, the answer is 11266.
>
> In this example, the model actually performs the addition for 442 + 684, which indicates a biased overfitting from the initialization phase. During the iteration phase, the skipping data helps to mitigate this overfitting, thereby improving accuracy.
>
> **[Q5]**
>
> On the task of Analog of Algebra, the original training set contains 5770 data. As the iteration proceeds, the number of mixed data points (Total) increases significantly, nearly tripling by the fifth iteration. Here, $i$ indicates the number of valid answers that skip $i$ step in the reasoning process.
>
> |         | Total | i=1 | i=2 |
> |---------|-------|--------|--------|
> |Initialization   | 5770  | 0      | 0      |
> |Iter 1   | 5945  | 153    | 22     |
> |Iter 2   | 7078  | 1050   | 258    |
> |Iter 3   | 10010 | 2895   | 1345   |
> |Iter 4   | 13744 | 4753   | 3221   |
> |Iter 5   | 15048 | 5255   | 4023   |
>
> *Table: The change in data volume across iterations.*
>
> **[Q6]**
>
> In the warm start phase, we only use i=1 in the manual skipping data.
>
> **[Q7]**
>
> Thank you for pointing it out. We will revise the figure reference accordingly.

---

> > ### Comment · Reviewer_DBKK · 2024-08-11
> >
> > Thank you for the thorough responses, they all make sense, and I appreciate the additional concrete evidence. I think adding/adjusting the metrics in your response to Q1 and Q3 on all tasks could really enhance the paper, and have raised my score accordingly.

---

> > > ### Author Response · Authors · 2024-08-14
> > >
> > > We are grateful for your recognition of our work! We here address the remaining concern as follows.
> > >
> > > **[W1] Generalization across different domains**
> > >
> > > We appreciate the insightful question regarding the generalization of step-skipping ability across different tasks. To address this concern, we conduct a controlled experiment. Specifically, we sample 2000 training examples from each dataset, along with 1600 step-skipping answers where exactly one step is successfully skipped, drawn from these 2000 questions in Iteration 5. This approach ensures that all three tasks have an equal amount of full-step and step-skipping data.
> > >
> > > We utilize the Phi-3-mini model across all tasks. In the table below, the "withheld task" refers to the task that does not have step-skipping data in the training phase. The "All" setting includes only full-step answers across all three tasks without any step-skipping data.
> > >
> > > To clarify the settings:
> > > * All setting = task1-full + task2-full + task3-full
> > > * Withheld setting = task1-full + *task1-skip* + task2-full + *task2-skip* + task3-full
> > >
> > > *Table 1: Evaluation task - Analog of Algebra*
> > >
> > > | Withheld task       | In-domain | Avg steps | OOD-easy | Avg steps | OOD-hard | Avg steps |
> > > |---------------------|-----------|-----------|----------|-----------|----------|-----------|
> > > | All                 | 51.3      | 2.65      | 44.5     | 5.58      | 1.9      | 10.68     |
> > > | Analog of Algebra   | 53.9      | 2.71      | 56.9     | 5.74      | 7.1      | 10.97     |
> > >
> > > *Table 2: Evaluation task - Multi-digit Addition*
> > >
> > > | Withheld task       | In-domain | Avg steps | OOD-easy | Avg steps | OOD-hard | Avg steps |
> > > |---------------------|-----------|-----------|----------|-----------|----------|-----------|
> > > | All                 | 100.0     | 2.86      | 22.4     | 4.71      | 4.2      | 5.39      |
> > > | Multi-digit Addition| 95.7      | 2.59      | 34.3     | 4.75      | 2.4      | 5.35      |
> > >
> > > *Table 3: Evaluation task - Directional Reasoning*
> > >
> > > | Withheld task          | In-domain | Avg steps | OOD-easy | Avg steps | OOD-hard | Avg steps |
> > > |------------------------|-----------|-----------|----------|-----------|----------|-----------|
> > > | All                    | 100.0     | 7.01      | 96.0     | 15.46     | 75.8     | 25.03     |
> > > | Directional Reasoning  | 97.8      | 6.98      | 96.2     | 15.42     | 80.0     | 24.92     |
> > >
> > > The results show that the presence of step-skipping data in the training of other tasks positively impacts the performance of the withheld task. Compared to the "All" setting, models trained with step-skipping data in other tasks demonstrate improved step-skipping performance across all datasets with a comparable number of steps. Since the model is explicitly instructed to solve problems with fewer steps, the average steps in the withheld task remain similar to the "All" setting. However, the observed boost in accuracy suggests that the model benefits from the generalized step-skipping ability acquired from other tasks. This confirms the generalizability of the step-skipping ability across different domains.
> > >
> > > We appreciate your insightful questions and believe that these findings will further strengthen our work. We will make sure to incorporate these valuable suggestions into our revision. Thank you again for taking the time to review our work thoroughly.

---

### Official Review · Reviewer_vxGY · 2024-07-13

**Soundness:** 3
**Presentation:** 3
**Contribution:** 3
**Rating:** 4
**Confidence:** 4

**Summary:**

This paper proposes to teach LLMs to deliberately skip steps when doing complex tasks involving multi-step reasoning. The authors use self-generated inference paths with fewer steps to fine-tune the models, which is similar to self-distillation. The authors conduct experiments on a few controlled tasks show that the proposed approach can effectively reduces the reasoning steps while maintain performance.

**Strengths:**

1. The idea of teaching LLMs to skip steps following the human reasoning process is intuitive and makes sense.
2. The proposed method is overall technically sound and well described.
3. The paper is in general well-written and easy to follow.
4. Experimental results confirm the effectiveness of the proposed approach, at least on these "artificial" tasks.

**Weaknesses:**

1. The experiments are not solid because the tasks considered in the experiments are very artificial and not representative for real-world reasoning tasks. The paper could be made much stronger by conducting tasks/datasets such as GSM8K/MATH or coding tasks, instead of simple reasoning tasks. Without the empirical study on realistic tasks, it is hard to confirm the contribution and usefulness of the proposed metric.

**Questions:**

N/A

**Limitations:**

Yes

---

> ### Author Rebuttal · Authors · 2024-08-06
>
> Thank you for your constructive feedback and valuable suggestion! We acknowledge the importance of using established practical benchmarks. We are currently conducting additional experiments with these datasets and will provide an update on the results once the experiments are completed.

---

### Official Review · Reviewer_rzDm · 2024-07-13

**Soundness:** 3
**Presentation:** 3
**Contribution:** 3
**Rating:** 7
**Confidence:** 4

**Summary:**

The paper proposes a method for training an LLM to solve reasoning problems using fewer verbalized reasoning steps than it is naturally encouraged to by a fixed training dataset. The resulting model is shown to maintian or improve performance on in-distribution data and OOD data testing extrapolation w.r.t. length or compositionality, while using fewer reasoning steps at inference time. Analysis shows that performance gains are concentrated around problems requiring an intermediate number of reasoning steps, rather than very few reasoning steps. Experiments are conducted with Llama-2-7b on three synthetic datasets. The method itself works by using warm-start data with mixed length reasoning demonstrations, followed by bootstrapped training data created by controlling model generations with control codes (instructions) combined with filtering model generations for correctness to create new gold data.

**Strengths:**

- Very important: The idea of shortening reasoning steps, particularly to mimic human reasoning that is variable in its verbalized length, is a very interesting and practical direction.
- Very important: Results are positive and promising for model generalization at an increased level of efficiency. Particularly interesting are results suggesting that model performance can increase on difficult OOD data by virtue of skipping some reasoning steps. Initially, CoT was found to improve OOD generalization, but it seems that this iteration on CoT could improve OOD performance even more in some situations.
- Important: The paper is overall straightforward to read and understand, with only a few exceptions.
- Of some importance: The connection to easy-to-hard generalization was interesting to me. That this method could improve OOD performance, specifically length/compositional generalization, was very interesting.

**Weaknesses:**

- Important: What are the instructions at inference time? Do you require a ground-truth number of reasoning steps to run the model at inference time? If so, this important detail is missing from the paper and could make the method difficult to use in practice for problems if it is not know how difficult they are in advance. Would the method be robust to misspecified instructions at inference time?
- Important: I find it a little confusing to reconcile the results of Sec. 5.1 with Sec. 5.2. Sec. 5.1 makes it look like using fewer steps greatly hurts model performance, while Sec. 5.2 makes it seem like using fewer steps does not hurt performance (specifically the Warm start rows, relative to Cold start baselines).
- Of some importance: The data is a little artifical. There are existing reasoning and compositional reasoning benchmarks that could be appropriate for this work (though they could require stronger models), including SCAN (https://arxiv.org/abs/1711.00350), GSM8k, StrategyQA, and MATH datasets. However, this is not a major weakness as using clean, controlled datasets is advantageous for studying these kinds of phenomena and they enable automatic construction of warm start data.

**Questions:**

- Why keep the cold start data in the training data if the bootstrapped data is good or better? Do you have ablations that suggest what mixture of the data is best?
- Suggested experiment: if you could have two models that are similar except for one being better at long-context reasoning, it would be interesting to see how your method affects each model. The reason for this is that compressing reasoning length could be beneficial by virtue of reducing the context length, rather than some other inherent benefit like allowing the model to spend more computation on harder steps. Such an experiment would help disambiguate if the improvement comes from shortening context length or from using fewer steps.
- Note L.68-69 is heavily disputed by follow work on ToM, e.g. https://arxiv.org/pdf/2310.19619
- Just so you’re aware, some highly related work has appeared contemporaneously: (1) https://arxiv.org/pdf/2405.14838, (2) https://arxiv.org/pdf/2407.06023
- L.34: use an em-dash rather than single dash here
- L.221: Fig7(a) should read Fig4(a)

**Limitations:**

Discussion was adequate, but it could be worth mentioning that experiments were conducted with only one model and three synthetic datasets.

---

> ### Author Rebuttal · Authors · 2024-08-06
>
> Thank you for your insightful and encouraging feedback. We are pleased that you found our approach and results promising and will continue to refine and expand upon these ideas in future revisions.
>
> **[W1]**
>
> We only require the step number as input when generating the skipped data for the training set during the iteration phase. At inference time, we train an additional standard model (described as Eq. 2), which only takes the question as input, without the need for a number of reasoning steps. The instruction used is:
>
> "Transform the expression to isolate ❤ on the left side of the ↔. \
> Question: [[QUESTION]] \
> Answer:"
>
> This approach ensures the model is robust and practical for real-world applications, as it does not rely on the ground-truth number of reasoning steps during inference.
>
> **[W2]**
>
> We apologize for the confusion regarding the results in Sections 5.1 and 5.2. We will provide more detailed captions to clarify this. In Section 5.1, we analyze the initialized model $M_0$, which is trained with full step data only and requires the step number as input. When asked to generate answers in fewer steps, the model struggles to maintain accuracy because it has not been trained on the skipping-step data. The primary purpose of $M_0$ is to generate skipped step data during the iteration phase, and it is not expected to achieve high accuracy initially—only sufficient accuracy to generate usable data.
>
> In Section 5.2, we evaluate the resulting standard model $M^{standard}$, which is trained on both the full step and the generated skipped data, and does not require the step number as input. This training method allows $M^{standard}$ to maintain accuracy while using fewer steps, demonstrating the effectiveness of our approach.
>
> **[W3]**
>
> Thank you for your valuable suggestion! We acknowledge the importance of using established benchmarks. We are actively working on additional experiments using these datasets and will provide an update on the results once the experiments are completed.
>
>
> **[Q1]**
>
> Mixing the data allows us to gradually process the iterations. At the beginning of the iterations, the valid skipping data may not be enough and they cannot help the model fully understand the task. For example, on Analog of Algebra, Iter 1 only has less than 150 valid skipping data to use. We are also working on additional ablation analysis to provide empirical support following your advice.
>
> **[Q2]**
>
> Thank you for this insightful suggestion! It would indeed be valuable to investigate where the improvements come from. We will consider incorporating this experiment in future revisions to better understand the underlying reasons.
>
> **[Q3]**
>
> Thank you for pointing this out. We will include this in the related work.
>
> **[Q4]**
>
> Thank you for providing the references. We are excited to see related work in the community that shares similar ideas and interests! We will incorporate these references in the revised version.
>
> **[Q5, Q6]**
>
> Thank you for pointing out these typos. We will revise them accordingly.
>
> **[Limitation]**
>
> We additionally conduct experiments with Phi-3-mini on the Analog of Algebra task. The results confirm that our method is generalizable to other model series. We will include the results across all tasks in the revision.
>
>
> |          | In-domain    |            | OOD-easy   |            | OOD-hard   |            |
> |----------|--------------|------------|------------|------------|------------|------------|
> |          | Acc          | Avg steps  | Acc        | Avg steps  | Acc        | Avg steps  |
> | Cold Start | 99.5       | 3.18       | 96.86      | 6.16       | 1.67       | 9.83       |
> | Iter 1   | 99.7         | 3.18       | 99.22      | 6.13       | 0.95       | 10.61      |
> | Iter 2   | 99.9         | 3.11       | 100.0        | 6.06       | 5.48       | 9.06       |
> | Iter 3   | 99.9         | 2.90        | 99.6       | 5.79       | 6.67       | 7.74       |
> | Iter 4   | 99.9         | 2.64       | 99.41      | 5.36       | 9.05       | 7.46       |
> | Iter 5   | 99.7         | 2.43       | 98.82      | 5.23       | 10.24      | 7.77       |
>
> Table: Phi-3-mini on Analog of Algebra.

---

> > ### Comment · Reviewer_rzDm · 2024-08-10
> > **Respose to rebuttal**
> >
> > >We only require the step number as input when generating the skipped data for the training set during the iteration phase. At inference time…
> >
> > Thanks for the clarification! This highlights to me that there may not be strong control over how much step skipping the model does. Rather, the model learns to skip some portion of the time based on the training data. I think this is fine. I think the strength of this paper is interesting results around step skipping and model generalization, and not a new method for controlling how many steps a model takes when solving a problem.
> >
> > >We apologize for the confusion regarding the results in Sections 5.1 and 5.2…
> >
> > Thanks this makes sense!
> >
> > >We are actively working on additional experiments using these datasets
> >
> > Great, please do upload these if they finish in time.
> >
> > ---
> >
> > Based on the above discussion, I plan on keeping my score at 7. If the results were especially interesting on additional, harder datasets, I could increase it, but I think the core contribution of the paper is already solid based on the three datasets used.

---

> > > ### Author Response · Authors · 2024-08-14
> > >
> > > Thank you so much for your recognition and support of our work!
> > >
> > > We have conducted additional experiments to evaluate generalization across different domains. Our findings indicate that step-skipping data from other tasks positively impacts the step-skipping performance on the withheld task. We kindly invite you to review the detailed results in our response to Reviewer DBKK and Reviewer E6Qj.
> > >
> > > We will make sure to incorporate these valuable suggestions into our revision. Once again, we sincerely appreciate the time and effort you’ve taken to thoroughly review our work.

---

### Decision · Program_Chairs · 2024-09-25

**Decision:**

Accept (poster)

**Comment:**

In this work, the authors explore how to guide LLM's training so they can skip unnecessary reasoning steps like humans do. Concretely, they propose a framework that iteratively refining models to generate shorter reasoning paths, at every iteration, they keep the generated shorter reasoning paths that lead to the correct answers to fine-tune the model in the next iteration. By doing so, they show that on three dataset, a llama2-based model could solve tasks with less steps but not sacrificing accuracy.

This work has many merits. All reviewers agree that the idea is interesting and intuitive, shortening reasoning steps could be very useful practically; the proposed method is technically sound; the paper is clearly written, the structure is well organized; the three tasks, despite synthetic, are well designed to show how and why the proposed method could work, the experimental results support well the authors' claim; the authors provide detailed analyses that helped readers to understand the training pipeline; the idea is generally well connected to the paradigm of easy-to-hard generalization.

On the negative side, reviewers' concerns were mainly around 1) clarity on some details; 2) the relatively simple datasets/tasks; and 3) the authors failed to demonstrate the method with more than one LLM backbones.

To address the concerns, during the author-reviewer discussion period, the authors provided the following additional information:
- The authors added experimental results with Phi-3-mini, in addition to the Llama2-7B in their submission. Results provide more evidence on generalizability of the method on other backbone models.
- The authors provided an analysis on the skipping ratio (the proportion of answers where steps were skipped in the entire test set) at every iteration. This shows that the model indeed learns to skip more steps (while accuracy keeps unchanged or even improved) with more training iterations.
- The authors provided an additional experiment testing the method's generalizability across different domains. They show that training the model with such step-skipping behavior could generalize to unseen tasks.
- The authors clarified some points that were unclear before.

The authors agreed to conduct experiments on more realistic reasoning datasets such as GSM8K, MATH, or coding. However, due to the limited time, they fail to finish that set of experiments (which is understandable). Because of this, Reviewer vxGY remains their negative score of 4, which makes sense, I also understand.

In general, the authors did a great job addressing the reviewers' concerns and answering their questions. Reviewer DBKK increased their score from 4 to 6. During the reviewer-AC discussion period, both Reviewer DBKK and rzDm involved in the discussion, they both express that while acknowledging that the work could be much stronger if they could include the more realistic datasets, they believe the current version (initial submission + authors' response) gives them enough confidence that the work could be well received by the NeurIPS community.

Personally, I agree with all reviewers, I think this work provides some interesting insights on how models could be guided to evolve from system 1 to system 2 thinking. At the same time, I also agree with Reviewer vxGY that more complex / non-synthetic datasets could be crucial in better justifying the method. I tentatively suggest an accept.